# Advances in Treatment of Brominated Hydrocarbons by Heterogeneous Catalytic Ozonation and Bromate Minimization

**DOI:** 10.3390/molecules24193450

**Published:** 2019-09-23

**Authors:** Asogan N. Gounden, Sreekantha B. Jonnalagadda

**Affiliations:** 1Department of Chemistry, Mangosuthu University of Technology, P.O. Box 12363, Jacobs 4026, South Africa; asogan@mut.ac.za; 2School of Chemistry, Westville Campus, University of KwaZulu-Natal, P Bag X54001, Durban 4000, South Africa

**Keywords:** bromate minimization, catalytic ozonation, metal oxides, bromide

## Abstract

The formation of carcinogenic bromate ions is a constraint when ozone is used for the remediation of water containing brominated organic materials. With its strong oxidizing ability, ozone rapidly transforms bromide in aqueous media to bromate, through a series of reactions involving hydroxyl radicals. Several strategies, such as limiting the ozone concentration, maintaining pH < 6, or the use of ammonia or hydrogen peroxide were explored to minimize bromate generation. However, most of the above strategies had a negative effect on the ozonation efficiency. The advanced oxidation processes, using catalysts together with ozone, have proven to be a promising technology for the degradation of pollutants in wastewater, but very few studies have been conducted to find ways to minimize bromate formation during this approach. The proposed article, therefore, presents a comprehensive review on recent advances in bromate reduction in water by catalytic ozonation and proposes reaction mechanisms associated with the catalytic process. The main aim is to highlight any gaps in the reported studies, thus creating a platform for future research and a quest to find environment friendly and efficacious catalysts for minimizing bromate formation in aqueous media during ozonation of brominated organic compounds.

## 1. Introduction

The need to reduce environmental pollution is currently receiving urgent attention around the world. The rapid increase in the human population, coupled with growing demands from industrial and other sectors, has triggered the large-scale usage of diverse non-biodegradable chemicals, leading to extensive pollution of water systems. Since these polluted waters pose a serious threat to the environment, ongoing research is conducted to explore cost effective treatment methodologies for the removal of varied toxic chemicals from the water systems. An alternative to chlorination and adsorption agents for water purification is ozonation, which is becoming a useful methodology for improving the quality of water. The use of ozone has proven to be excellent for microorganism destruction and biological contaminant removal from water [1], but is not effective for degrading recalcitrant organic pollutants in water. The presence of bromide $\left({\mathrm{Br}}^{-}\right)$ in polluted waters poses a serious problem during ozonation. Bromide is rapidly oxidized to toxic bromate $\left({\mathrm{BrO}}_{3}^{-}\right)$ during ozone treatment. Bromide is usually present in low concentrations of between 10^4^ and 10^6^ ppb in wastewaters and approximately 67 × 10^3^ ppb in seawater [2]. Relatively low amounts were found in rainwater, ranging from 0 to 110 ppb [3], but in groundwater, between 10 and 2 × 10^3^ ppb were detected [4]. Higher bromide concentrations have been reported in waters and soil samples near oceans [5]. Mining and leaded petrol [6], fertilizers and insecticides are considered major sources of bromine contamination of the environment and aquatic systems [7]. Bromide was also found in many treated water facilities ranging from 3000 to 10,000 ppb [8,9]. If bromide levels as low as 20 ppm are present in water during ozonation, the potential exists for bromate formation to occur through a combination of ozone and hydroxyl radical reactions [1]. Bromate is a known human carcinogen [10,11,12] and its maximum allowable limit in drinking water is set at 10 ppb or lower [13]. Therefore, it is crucial to minimize or prevent its formation in drinking water. 

In aqueous systems, ozone oxidizes bromide to bromate via three different pathways [14]. The dominance of a particular pathway is dependent on the amount of bromide, organic carbon and pH of the substrate solution. As illustrated in Figure 1, the first pathway (direct pathway) is initiated by the reaction of bromide ion with molecular ozone to form ${\mathrm{OBr}}^{-}/\mathrm{HOBr}$. The ${\mathrm{OBr}}^{-}$ is further oxidized by dissolved $O_{3}$ to ${\mathrm{BrO}}_{2}^{-}$ and finally to ${\mathrm{BrO}}_{3}^{-}$. The second pathway (direct/indirect pathway) is facilitated by the molecular ozone, resulting in the formation of ${\mathrm{OBr}}^{-}/\mathrm{HOBr}$. However, in this route the formed ${\mathrm{OBr}}^{-}$ is oxidized by ${\mathrm{HO}}^{•}$ radicals to a series of highly reactive oxygenated radicals. Further ozonation produces ${\mathrm{BrO}}_{3}^{-}$ ions. According to Richardson et al. [15], this pathway is favoured if solution pH and alkalinity of the water is high. In the third pathway, the ${\mathrm{HO}}^{•}$ radicals interact with bromide ions resulting in the generation of BrO radicals, which is disproportionate to bromite ions. The bromite ions are then oxidized by molecular ozone to produce bromate ions. 

The use of suitable heterogeneous catalysts has proven to be beneficial to enhance the efficiency of the ozonation process and minimize the generation of toxic by-products [16]. Studies have shown that hydroxyl radicals generated during ozonation in the presence of metal oxides could increase bromate formation [17]. This review presents a comprehensive assessment on recent advances on bromate reduction in water by heterogeneous catalytic ozonation.

## 2. Bromate Minimization Strategies

The following mechanism was proposed by von Gunten and Hoigne’ for the conversion of ${\mathrm{Br}}^{-}$ to ${\mathrm{BrO}}_{3}^{-}$ during ozonation [18]: $${\mathrm{Br}}^{-}\to \;\mathrm{HOBr}/{\mathrm{OBr}}^{-}\to {\mathrm{BrO}}^{•}\to {\mathrm{BrO}}_{2}^{-}\to {\mathrm{BrO}}_{3}^{-}$$

They concluded that the direct oxidative conversion of ${\mathrm{Br}}^{-}$ to $\mathrm{HOBr}/{\mathrm{OBr}}^{-}$ was mainly controlled by molecular ozone, while further oxidation of $\mathrm{HOBr}/{\mathrm{OBr}}^{-}$ to ${\mathrm{BrO}}^{•}$ radicals was influenced by ${\mathrm{HO}}^{•}$ radicals. The unstable ${\mathrm{BrO}}^{•}$ radicals disproportionate to ${\mathrm{BrO}}_{2}^{-}$. The dissolved ozone in the water then rapidly oxidizes ${\mathrm{BrO}}_{2}^{-}$ to ${\mathrm{BrO}}_{3}^{-}$ [1]. Limited studies have been conducted to establish the effects of catalytic ozonation on bromate formation. The most recent studies are discussed below.

### 2.1. MCM-48, CeO_2_ and Ce_x_-MCM-48

Li et al. [19] reported on catalytic ozonation of bromide containing waters with MCM-48, CeO_2_ and combined mesoporous sieve Ce_x_-MCM-48 (cerium combined with MCM-48) with various Si/Ce molar ratios (Ce_30/66/100/200_-MCM-48). All catalysts were able to considerably impede ${\mathrm{BrO}}_{3}^{-}$ formation in comparison to ozonation alone. After 30 min of ozone treatment, the inhibition efficiencies of MCM-48 and CeO_2_ were 78.6% and 63.9%, respectively. When MCM-48 was doped with Ce, a marked improvement in ${\mathrm{BrO}}_{3}^{-}$ minimization was observed. When the Ce content was increased from x = 200 to x = 66, ${\mathrm{BrO}}_{3}^{-}$ yield decreased, giving a maximum inhibition efficiency of 91% after 30 min of ozonation. However, an additional increase of Ce to x = 30, resulted in an increase in ${\mathrm{BrO}}_{3}^{-}$ concentration and an inhibition efficiency of 78%. Their explanation for this trend was that doping MCM-48 with Ce resulted in the generation of more surface hydroxyl groups, which successively enhanced decomposition of O_3_ on the active sites of the catalyst surface. However, doping beyond x = 66 blocked the active sites, leading to a destruction of the mesoporous structure of MCM-48, hence leading to poor catalyst activity. 

Li et al. [19] proposed a bromate reduction pathway for Ce_66_-MCM-48 with the aid of bromine mass balance studies. Their results revealed that Ce_66_-MCM-48 did not adsorb ${\mathrm{Br}}^{-},\;\mathrm{HOBr}/{\mathrm{OBr}}^{-}$ and ${\mathrm{BrO}}_{3}^{-}$, the main bromine-containing species present in the water solution. The amounts of both $\mathrm{HOBr}/{\mathrm{OBr}}^{-}$ and ${\;\mathrm{BrO}}_{3}^{-}$ in Ce_66_-MCM-48 ozonation were expressively lower, relative to ozone in absence of catalyst, while the amount of ${\mathrm{Br}}^{-}$ was much higher in Ce_66_-MCM-48 ozonation. As the bromide oxidation is primarily controlled by $O_{3}$, Ce_66_-MCM-48 ozonation tends to prevent ${\mathrm{BrO}}_{3}^{-}$ production by limiting the influence of direct $O_{3}$ oxidation. The results have shown that $O_{3}$ decomposed faster with Ce_66_-MCM-48 (82% decomposition after 5 min), in comparison to ozonation alone (53% decomposition in the first 5 min). Since a lower amount of dissolved $O_{3}$ exists in Ce_66_-MCM-48 ozonation, the consecutive oxidation reactions from ${\mathrm{Br}}^{-}\to \mathrm{HOBr}/{\mathrm{OBr}}^{-}\to {\;\mathrm{BrO}}_{3}^{-}$ are all inhibited. The generated secondary oxidant, ${\mathrm{HO}}^{•},$ reacts with some bromine containing species, organic micropollutants, or combine to form $H_{2}O_{2}$. The results showed that $H_{2}O_{2}$ concentration steadily increases during ozonation alone, reaching a maximum value of 0.6 μM after 20 min. With ozonation in the presence of Ce_66_-MCM-48, a higher $H_{2}O_{2}$ concentration was detected, but it remained constant (1.5–1.7 μM) for the entire 20 min. Another bromate inhibition mechanism involved electron transfer reactions between ${\mathrm{Ce}}^{3+}$ and ${\mathrm{Ce}}^{4+}$ on Ce_66_-MCM-48 surface. These reactions lead to the inhibition of ${\mathrm{Br}}^{-}$ to $\mathrm{HOBr}/{\mathrm{OBr}}^{-}$, thus resulting in lower ${\mathrm{BrO}}_{3}^{-}$ formation. The ${\mathrm{Ce}}^{3+}$ surface ions underwent oxidation by ${\mathrm{Br}}^{•}$ and ${\mathrm{BrO}}^{•}$ to form ${\mathrm{Ce}}^{4+}$ [20] according to the following pathway:(1)$${\mathrm{Ce}}^{3+}+{\mathrm{BrO}}^{•}+H^{+}\;\to {\;\mathrm{Ce}}^{4+}+\mathrm{HOBr}\;$$
(2)$${\mathrm{Ce}}^{3+}+{\mathrm{Br}}^{•}\;\to {\;\mathrm{Ce}}^{4+}+{\mathrm{Br}}^{-}$$
Ce^3+^ also reacts with $H_{2}O_{2}$ to form ${\mathrm{Ce}}^{4+}$ [21]:(3)$${\mathrm{Ce}}^{3+}+H_{2}O_{2}+H^{+}\;\to {\;\mathrm{Ce}}^{4+}+{\mathrm{HO}}^{•}+H_{2}O\;$$
An alternative pathway produces $H_{2}O_{2}$ from aqueous $O_{3}$ decomposition
(4)$$O_{3}+{\mathrm{OH}}^{-}\;\to {\;\mathrm{HO}}_{2}^{-}+O_{2}$$
(5)$$O_{3}+{\mathrm{HO}}_{2}^{-}\;\to {\;\mathrm{HO}}^{•}+O_{2}^{•-}+O_{2}$$
(6)$${\mathrm{HO}}^{•}+{\mathrm{HO}}^{•}\;\to {\;H}_{2}O_{2}$$
(7)$$H_{2}O_{2}\;\to {\;\mathrm{HO}}_{2}^{-}+H^{+}$$
Ce^3+^ is regenerated by ${\mathrm{HO}}_{2}^{-}$, which converts ${\mathrm{Ce}}^{4+}$ to ${\mathrm{Ce}}^{3+}$ [21]:(8)$${\mathrm{Ce}}^{4+}+{\mathrm{HO}}_{2}^{-}\;\to {\;O}_{2}+{\mathrm{Ce}}^{3+}+H^{+}$$

### 2.2. α-FeOOH, α-Fe_2_O_3_, γ-FeOOH and CeO_2_

T. Zang et al. [22] investigated the effect of a number of metal oxides, such as α-FeOOH, $\alpha -{\mathrm{Fe}}_{2}O_{3}$, γ-FeOOH and ${\mathrm{CeO}}_{2,}$ on bromate production during ozone treatment of bromide in water. The catalytic reactions with $\alpha -{\mathrm{Fe}}_{2}O_{3}$ produced more ${\mathrm{BrO}}_{3}^{-}$ relative to ozonation alone, whereas the reactions with α-FeOOH, γ-FeOOH and ${\mathrm{CeO}}_{2}$ minimized bromate formation. However, ${\mathrm{CeO}}_{2}$ was most active in reducing bromate production. They determined simultaneously the concentrations of ${\mathrm{Br}}^{-}$ and $\mathrm{HOBr}/{\mathrm{OBr}}^{-}$ for uncatalysed ozonation and CeO_2_ catalysed ozonation. They found that the ${\mathrm{Br}}^{-}$ amounts in catalytic ozonation was lower with ozone treatment alone before 15 min, and remained similar thereafter. The $\mathrm{HOBr}/{\mathrm{OBr}}^{-}$ amount in ${\mathrm{CeO}}_{2}$ catalytic ozonation was always significantly higher in comparison to ozone treatment alone. According to von Gunten [1], $\mathrm{HOBr}/{\mathrm{OBr}}^{-}$ is an essential intermediary for ${\mathrm{BrO}}_{3}^{-}$ production during ozonation, therefore, its accumulation in ${\mathrm{CeO}}_{2}$ catalytic ozonation suggests that ${\;\mathrm{CeO}}_{2}$ considerably inhibits the conversion of $\mathrm{HOBr}/{\mathrm{OBr}}^{-}$ to ${\mathrm{BrO}}_{3}^{-}$.

The formation of $H_{2}O_{2}$ was detected in both ozonation alone and ozonation with ${\;\mathrm{CeO}}_{2}$. The results showed that the amount of $H_{2}O_{2}$ with ${\mathrm{CeO}}_{2}$ was poorer compared to single ozonation. Studies have shown that the surface of ${\;\mathrm{CeO}}_{2}$ can initiate the decomposition${\;H}_{2}O_{2}$ generating oxygen in water [23]. Therefore, the lesser $H_{2}O_{2}$ amount in ${\mathrm{CeO}}_{2}$ catalytic ozonation can be attributed to its concurrent disintegration on the surface of$\;{\mathrm{CeO}}_{2}$. One study mentioned that low amounts of hydrogen peroxide can promote ${\mathrm{BrO}}_{3}^{-}$ formation, arising from hydroxyl radical formation from the interaction of ${\mathrm{HO}}_{2}^{-}$ with $O_{3}$ [1], and other studies discussed that hydrogen peroxide at high amounts ($H_{2}O_{2}/O_{3}$ molar ratio >1:2) is likely to reduce $\mathrm{HOBr}/{\mathrm{OBr}}^{-}$ to ${\mathrm{Br}}^{-}$, hence minimizing ${\mathrm{BrO}}_{3}^{-}$ formation [17,18,24]. According to Zang et al. [22], the enhanced ${\mathrm{BrO}}_{3}^{-}$ minimization in ${\mathrm{CeO}}_{2}$ catalytic ozonation is primarily due to the lower $H_{2}O_{2}$ amounts. Since ${\mathrm{CeO}}_{2}$ catalytic ozonation produced a lower amount of $H_{2}O_{2}$ than single ozonation, the$\;{\mathrm{HO}}^{•}$ amount is expected to be moderately lower, hence resulting in a lower oxidation rate of $\mathrm{HOBr}/{\mathrm{OBr}}^{-}$ to ${\mathrm{BrO}}^{•}$. Furthermore, ${\mathrm{BrO}}^{•}$ can be reduced to $\mathrm{HOBr}/{\mathrm{OBr}}^{-}$ by ${\mathrm{Ce}}^{3+}$, which is a temporary reductive state of surface ${\mathrm{Ce}}^{4+}$ in catalytic decomposition of $H_{2}O_{2}$ [25]. Thus, an additional pathway for ${\mathrm{BrO}}_{3}^{-}$ minimization is the reduction of ${\mathrm{BrO}}^{•}$ to $\mathrm{HOBr}/{\mathrm{OBr}}^{-}$ on the ${\mathrm{CeO}}_{2}$ surface. Both ${\mathrm{BrO}}_{3}^{-}$ reduction routes require the involvement of surface active ${\mathrm{Ce}}^{4+}$ sites.

It has been reported that ${\mathrm{SO}}_{4}^{2-}$ ions, when combined with metal oxides, have a strong attraction for their surface sites [26]. Zang et al. [22], therefore, added various concentrations of ${\mathrm{SO}}_{4}^{2-}$ to the bromide containing solutions to ascertain its affinity for surface active ${\mathrm{Ce}}^{4+}$ sites, and the impact on ${\mathrm{BrO}}_{3}^{-}$. They found that the difference in bromate formation between ozonation alone and ${\mathrm{CeO}}_{2}$ catalytic ozonation decreased as${\;\mathrm{SO}}_{4}^{2-}$ amounts increased from 0 to 5 mM. The diminishing effectiveness of ${\;\mathrm{CeO}}_{2}$ to minimize ${\mathrm{BrO}}_{3}^{-}$ formation is ascribed to surface ${\mathrm{Ce}}^{4+}-{\mathrm{SO}}_{4}^{2-}$ co-ordination, thus indicating that surface ${\mathrm{Ce}}^{4+}$ sites account for most of the ${\mathrm{BrO}}_{3}^{-}$ minimization during ${\mathrm{CeO}}_{2}$ catalytic ozonation. 

### 2.3. Nano-Metal Oxides, SnO_2_ and TiO_2_

Wu et al. [27] conducted simulation studies to investigate the influence of nano-metal oxides, ${\mathrm{SnO}}_{2}$ and ${\mathrm{TiO}}_{2}$ on bromate generation in pure water during ozone treatment. Their results showed that ozonation in the presence of nano-metal oxides (${\mathrm{SnO}}_{2}$ and ${\mathrm{TiO}}_{2}$) as catalysts, minimized ${\mathrm{BrO}}_{3}^{-}$ generation to a greater extent, compared to single ozonation. However, nano-${\mathrm{TiO}}_{2}$ was most effective in inhibiting ${\mathrm{BrO}}_{3}^{-}$ formation. The experimental results showed that the concentrations of residual O_3_ and $\mathrm{HOBr}/{\mathrm{OBr}}^{-}$ were significantly lesser in nano-TiO_2_ catalysed ozonation relative to uncatalysed ozonation and nano-SnO_2_ ozonation, indicating that catalytic ozonation with nano-TiO_2_ decomposes more O_3_ to ${\mathrm{HO}}^{•}$ radicals. The lower ozone concentration results in lower$\;\mathrm{HOBr}/{\mathrm{OBr}}^{-}$, hence minimizing ${\mathrm{BrO}}_{3}^{-}$ formation. Furthermore, ${\mathrm{HO}}^{•}$ radicals can rapidly combine to generate $H_{2}O_{2}$, which can reduce $\mathrm{HOBr}/{\mathrm{OBr}}^{-}$ to ${\mathrm{Br}}^{-}$ [28,29]. The presence of humic acid influenced bromate generation. Increasing the humic acid concentration from 0 to 3.0 ppm resulted in a decrease in bromate formation. Humic acid reacts readily with O_3_ and hydroxyl radicals, which also reacts with ${\;\mathrm{Br}}^{-}$ and $\mathrm{HOBr}/{\mathrm{OBr}}^{-}$ [16,30]. Therefore, a lower concentration of $\mathrm{HOBr}/{\mathrm{OBr}}^{-}$ leads to lesser bromate formation [22]. 

### 2.4. Mn Incorporated MCM-41

Xue et al. [31] employed mesoporous Mn incorporated MCM-41 to hinder bromate production during catalytic ozonation of waters containing bromide. A comparison of the three temperature ramping rates (0.5 K min^−1^, 1 K min^−1^ and 2 K min^−1^) during calcination of Mn_X_-MCM-41 (X = 40, 80, 100 and 120, the molar ratio of Si/Mn), revealed that Mn_100_-MCM-41 with ramping rate of 1 K min^−1^ showed superior surface characteristics and the greatest bromate inhibition efficiency. A 96.7% inhibition efficiency was achieved after 60 min when compared to ozonation alone. XPS data revealed that Mn_100_-MCM-41 (1 K min^−1^) has more oxygen vacancies, which has tendency to adsorb and dissociate ${\;H}_{2}O$ to surface active species [32]. Ozone readily reacts with these surface-active species, resulting in less ozone exposure for ${\mathrm{Br}}^{-}$ oxidation to ${\mathrm{HOBr}/\mathrm{OBr}}^{-}$, hence minimizing bromate formation. The higher fraction of ${\mathrm{Mn}}^{2+}\;$ and ${\mathrm{Mn}}^{3+}$ in Mn-MCM-41 enhanced bromate inhibition efficiency.

Xue et al. revealed that the concentration of $\mathrm{HOBr}/{\mathrm{OBr}}^{-}$ during Mn_100_-MCM-41 ozonation was lower than single ozonation. They explained that Mn_100_-MCM-41 adsorbs $H_{2}O$ and dissociates to form surface active species. Ozone then readily reacts with these surface-active species, hence leading to low ozone exposure for ${\mathrm{Br}}^{-}$ oxidation $\mathrm{HOBr}/{\mathrm{OBr}}^{-}$. Furthermore, hydrogen peroxide was detected in both uncatalysed and Mn_100_-MCM-41 catalysed ozonation. The concentration of $H_{2}O_{2}$ increased steadily in Mn_100_-MCM-41 ozonation, but decreased in uncatalysed ozonation, signifying that more reactive oxygen species [32] is formed in the presence of Mn_100_-MCM-41. These species are capable of consuming ${\mathrm{HOBr}/\mathrm{OBr}}^{-}$ and preventing bromate formation. To verify the role of hydroxyl radicals, TBA (a potential ${\mathrm{HO}}^{•}$ radical scavenger) was introduced in both single ozonation and Mn_100_-MCM-41 ozonation. The bromate yield decreased for both processes, thus confirming that ${\mathrm{HO}}^{•}$ was primarily responsible for ${\mathrm{BrO}}_{3}^{-}$ production. In ozonation alone, the decrease in bromate yield is mainly attributed to the decrease in hydroxyl radicals. In Mn_100_-MCM-41/O_3_ process, the decreased bromate yield is due to the decrease in both hydroxyl radicals and residual ozone. A similar phenomenon was evident with Fe-Cu-MCM-41 [33].

### 2.5. Fe-MCM-41, Cu-MCM-41 and Fe-Cu-MCM-41

Chen et al. [33] showed that ozonation with Fe-MCM-41, Cu-MCM-41 and Fe-Cu-MCM-41 catalysts considerably reduced${\;\mathrm{BrO}}_{3}^{-}$ formation. The inhibition activity and bromate yield were as follows: Cu-MCM-41 (28.8 ppb) ≈ Fe-MCM-41 (31.5 ppb) > Fe-Cu-MCM-41 (124.5 ppb) > O_3_ (432.5 ppb). They attributed the bromate reduction to ozone decomposition by the catalysts, resulting in a reduced amount of ozone for bromate generation [19]. The higher bromate yield in Fe-Cu-MCM-41/O_3_ than in Fe-MCM-41/O_3_ and Cu-MCM-41/O_3_ systems, is due to more ${\mathrm{HO}}^{•}$ presence in the Fe-Cu-MCM-41/O_3_ system. The presence of both the redox couples, ${\mathrm{Fe}}^{3+}/{\mathrm{Fe}}^{2+}$ and ${\mathrm{Cu}}^{2+}/{\mathrm{Cu}}^{+}$ on the catalyst surface (confirmed by XPS analysis) further accelerated ozone decomposition into$\;{\mathrm{HO}}^{•}$ radicals. As illustrated in Figure 2, bromate is produced through both the direct and indirect oxidation of ${\mathrm{Br}}^{-}$ by $O_{3}{/\mathrm{HO}}^{•}$ [34]. 

After the addition of the catalyst, more ozone is consumed, resulting in a hindrance of the direct oxidation of ${\mathrm{Br}}^{-}$ to ${\mathrm{HBrO}/\mathrm{BrO}}^{-}$ by ozone (a key intermediate reaction for bromate generation), and additional oxidation of ${\mathrm{HBrO}/\mathrm{BrO}}^{-}$ to ${\mathrm{BrO}}_{3}^{-}$ [19]. The superior efficiency of Fe-Cu-MCM-41, causes an abundance of hydroxyl radicals. A greater ${\mathrm{HO}}^{•}\;$ concentration results in an impediment of pathway 1, thus resulting in a higher bromate build-up [35].

The addition of t-butanol (TBA) to the${\;\mathrm{Br}}^{-}$ substrate solution, generated less bromate in both single ozonation and ozonation with Fe-Cu-MCM-41. As reported, the bromate formation requires the presence of both ozone and hydroxyl radicals [36]. Bromide is first oxidized by ozone directly to ${\mathrm{HBrO}/\mathrm{BrO}}^{-}$. Thereafter, the${\;\mathrm{HBrO}/\mathrm{BrO}}^{-}$ is oxidized by ${\mathrm{HO}}^{•}$ to ${\mathrm{BrO}}_{3}^{-}$. Thus, in single ozonation, since the ${\mathrm{HO}}^{•}$ radicals are scavenged by TBA, bromate formation is primarily due to molecular ozone. In the Fe-Cu-MCM-41/O_3_ process, the ozone concentration in the water significantly decreases due to the surface reactions, and the generated ${\mathrm{HO}}^{•}$ radicals are also scavenged by TBA. Both actions result in the suppression of the bromate formation pathway, hence, lowering bromate yield.

Bromate production was also inhibited in both ozonation alone and Fe-Cu-MCM-41 catalytic ozonation with the addition of ${\mathrm{PO}}_{4}^{3-}$. Bromide yields were found to increase with an increase in ${\mathrm{PO}}_{4}^{3-}$ dosage. As proposed by Huang, ${\mathrm{PO}}_{4}^{3-}$ accelerates the generation of $H_{2}O_{2}$, which reduces ${\mathrm{HBrO}/\mathrm{BrO}}^{-}$ to ${\mathrm{Br}}^{-}$, hence constraining ${\mathrm{BrO}}_{3}^{-}$ generation [37]. 

### 2.6. Fe–Al LDH Supported on Mesoporous Al_2_O_3_

Nie et al. [38] prepared Fe–Al layered double hydroxides (Fe-Al LDH, the molar ratio of ${\mathrm{Fe}}^{2+}$:${\mathrm{Fe}}^{3+}$ = 1:10) supported on mesoporous ${\mathrm{Al}}_{2}O_{3}$ and showed its effectiveness to minimize bromate formation. The ${\mathrm{BrO}}_{3}^{-}$ concentration rapidly increased during the uncatalysed ozonation reaching 20 ppb after 60 min of ozone treatment. However, ozonation with Fe-Al LDH/Al_2_O_3_ completely inhibited ${\mathrm{BrO}}_{3}^{-}$ formation. Furthermore, even when the initial ${\mathrm{Br}}^{-}$ concentration and ozone dose were increased, the ${\mathrm{BrO}}_{3}^{-}$ yield after 60 min of catalytic ozonation stayed below the allowable limit of 10 ppb. 

Fe-Al LDH/Al_2_O_3_ in the presence of a mixture of phenazone (PZ) and ${\mathrm{BrO}}_{3}^{-}$ only, revealed that approximately 45% of ${\mathrm{BrO}}_{3}^{-}$ was adsorbed on Fe-Al LDH/Al_2_O_3_ and 18% of ${\mathrm{Br}}^{-}$ was generated. They ascribed the ${\mathrm{BrO}}_{3}^{-}$ reduction to ${\mathrm{Fe}}^{2+}$ formed during Fe-Al LDH/Al_2_O_3_ preparation, which was confirmed by XPS analysis [39]. However, 82% of ${\mathrm{BrO}}_{3}^{-}$ was converted to ${\mathrm{Br}}^{-}$ during Fe-Al LDH/Al_2_O_3_ ozonation of the PZ/${\mathrm{BrO}}_{3}^{-}$ mixture. The reduction of ${\mathrm{BrO}}_{3}^{-}$ to ${\mathrm{Br}}^{-}$ increased with the ozone dose and ${\mathrm{BrO}}_{3}^{-}$ concentration. In contrast, the PZ/O_3_ system could not reduce ${\mathrm{BrO}}_{3}^{-}$ to ${\mathrm{Br}}^{-}$. Furthermore, when phosphate was added to the Fe-Al LDH/Al_2_O_3_/O_3_ system, ${\mathrm{BrO}}_{3}^{-}$ reduction was completely suppressed. The presence of phosphate permanently blocked the active surface sites of the catalyst, resulting in the replacement of surface hydroxyl groups and the formation of complexes with ${\mathrm{Fe}}^{3+}$ within the catalyst, thereby decreasing catalytic activity [40,41]. The adsorption of ${\mathrm{BrO}}_{3}^{-}$ and the interaction of O_3_ with Fe-Al LDH/Al_2_O_3_ was suppressed, therefore, poor ${\mathrm{BrO}}_{3}^{-}$ reduction is expected. Further investigations indicated that ${\mathrm{BrO}}_{3}^{-}$ reduction to ${\mathrm{Br}}^{-}$ by surface ${\mathrm{Fe}}^{2+}$ is responsible for complete inhibition of ${\mathrm{BrO}}_{3}^{-}$ formation. The ${\mathrm{Fe}}^{2+}$ needed for ${\mathrm{BrO}}_{3}^{-}$ reduction is generated from surface reactions occurring on Fe-Al LDH/Al_2_O_3_. The ${\mathrm{Fe}}^{3+}$- intermediate complex on the catalyst surface undergoes electron transfer reactions to produce ${\mathrm{Fe}}^{2+}$. Furthermore, the reaction of ${\mathrm{Fe}}^{3+}$ with ${\mathrm{HO}}_{2}^{•-}/O_{2}^{•-}$ forms ${\mathrm{Fe}}^{2+}$. The results also revealed that bromate reduction was favoured in the presence of different organic pollutants during catalytic ozonation. The amount of surface ${\mathrm{Fe}}^{2+}$, confirmed by XPS analysis, on Fe-Al LDH/Al_2_O_3_ varied for different organic pollutants, suggesting that the structure of the organic pollutant had an impact on the reduction of ${\mathrm{BrO}}_{3}^{-}$.

### 2.7. Mesoporous Alumina Supported MnOx

Nie et al. [42] investigated the reduction pathway of ${\mathrm{BrO}}_{3}^{-}$ generation during ozonation of 2,4-dichlorophenoxyacetic acid (2,4-D) with mesoporous alumina supported MnOx (MnOx/Al_2_O_3_) suspension. The ozonation of 2,4-D in the presence of bromide resulted in a rapid increase in bromate yield. The degradation of 2,4-D was significantly suppressed, while the efficiency of TOC removal decreased significantly from 25.7% to 7%. The catalytic ozonation with MnOx/Al_2_O_3_ significantly inhibited ${\mathrm{BrO}}_{3}^{-}$ formation, however, the presence of ${\mathrm{Br}}^{-}$ did not influence 2,4-D degradation.

In agreement with other studies, ${\mathrm{HBrO}/\mathrm{BrO}}^{-}$ was found to be the main essential intermediate for ${\mathrm{BrO}}_{3}^{-}$ formation [18]. During both the uncatalysed and catalysed ozonation, ${\mathrm{HBrO}/\mathrm{BrO}}^{-}$ was rapidly generated. However, ${\mathrm{BrO}}_{3}^{-}$ generation was significantly supressed with MnOx/Al_2_O_3_ in comparison to single ozonation. The trend in the data suggested that different bromine transformation mechanisms existed in the two processes. Bromate reduction occurred over MnOx/Al_2_O_3_ with ozone and 2,4-D, while a rapid increase in ${\mathrm{Br}}^{-}$ yield was observed. The results confirmed that ${\mathrm{BrO}}_{3}^{-}$ was reduced to ${\mathrm{Br}}^{-}$ on the surface of MnOx/Al_2_O_3_ during ozonation. Electron transfer reactions occurred during the $O_{3}$ adsorption and decomposition processes on the surface of the catalyst [43,44,45]. The UV–Vis absorption spectrum of MnOx showed the existence of Mn in different oxidation states, namely $Mn^{2+}$, $Mn^{3+}$ and $Mn^{4+}$ [46]. Therefore, $Mn^{2+}$ is responsible for promoting $O_{3}$ to eliminate organic pollutants and also assist in inhibiting ${\mathrm{BrO}}_{3}^{-}$ formation. The proposed reactions on MnOx/Al_2_O_3_ in the presence of ozone occurs as follows [42]: (9)$$O_{3}+{\mathrm{OH}}^{-}\;\to {\;O}_{2}^{•-}+{\mathrm{HO}}_{2}^{•}$$
(10)$${\mathrm{Mn}}^{4+}+O_{2}^{•-}\;\to {\;\mathrm{Mn}}^{3+}+O_{2}$$
(11)$${\mathrm{Mn}}^{3+}+O_{2}^{•-}\;\to {\;\mathrm{Mn}}^{2+}+O_{2}$$
(12)$${\mathrm{BrO}}_{3}^{-}+{\mathrm{Mn}}^{2+}\;\to {\;\mathrm{Br}}^{-}+{\mathrm{Mn}}^{3+}/{\mathrm{Mn}}^{4+}$$
(13)$$\mathrm{HBrO}/{\mathrm{BrO}}^{-}+{\mathrm{Mn}}^{2+}\;\to {\;\mathrm{Br}}^{-}+{\mathrm{Mn}}^{3+}/{\mathrm{Mn}}^{4+}$$
(14)$${\mathrm{HO}}_{2}^{•}+{\mathrm{HO}}_{2}^{•}\;\to {\;O}_{2}+H_{2}O_{2}$$

Reaction (14) proposes the generation of $H_{2}O_{2}$ in both uncatalysed and catalytic ozonation. The results showed that $H_{2}O_{2}$ concentration was remarkably lower in uncatalysed ozonation than in MnOx/Al_2_O_3_ catalytic ozonation. This trend suggests that in catalytic ozonation, reaction (14) is suppressed, since more ${\mathrm{HO}}_{2}^{•}$ is used up by reactions (10) and (11), hence leading to increased generation of ${\mathrm{Mn}}^{2+}$. This confirmed that the presence of different oxidation states of manganese is responsible for controlling ${\mathrm{BrO}}_{3}^{-}$ generation. 

### 2.8. Ce_x_ Zr_x-1_O_2_ Mixed Oxides

Yang et al. [47] prepared mixed oxides ${\mathrm{Ce}}_{x}{\mathrm{Zr}}_{x-1}O_{2}$ (x = 0.16, 0.50, 0.75, 0.9) and ${\mathrm{CeO}}_{2}$ to study ${\mathrm{BrO}}_{3}^{-}$ reduction during ozonation of ${\mathrm{Br}}^{-}$ containing filtered water. The results indicated that catalytic ozonation with ${\mathrm{Ce}}_{x}{\mathrm{Zr}}_{x-1}O_{2}$ and ${\mathrm{CeO}}_{2}$ minimized bromate formation better than ozonation alone. They concluded that the ${\mathrm{Ce}}_{x}{\mathrm{Zr}}_{x-1}O_{2}$ mixed oxides and ${\mathrm{CeO}}_{2}$ effectively suppressed the oxidation of ${\mathrm{Br}}^{-}$ by $O_{3}$ and ${\mathrm{HO}}^{•}$ radicals. Furthermore, the ${\mathrm{Ce}}_{0.75}{\mathrm{Zr}}_{0.25}O_{2}$ mixed oxide displayed the best catalytic activity for ${\mathrm{BrO}}_{3}^{-}$ minimization, with 53% of ${\mathrm{BrO}}_{3}^{-}$ formation being reduced after 20 min of ozonation. The adsorption of ${\mathrm{Br}}^{-}$ and ${\mathrm{BrO}}_{3}^{-}$ on catalyst surface were not detected, since anions have no affinity for the neutral or negatively charged oxide surface. Furthermore, the catalyst material exhibited good stability, since no leaching of metal ions were detected during the ozonation process. 

To confirm the role of $O_{3}$ and ${\mathrm{HO}}^{•}$ radicals in ${\mathrm{BrO}}_{3}^{-}$ inhibition, p-chlorobenzoic acid (pCBA), a ${\mathrm{HO}}^{•}$ scavenger was introduced to monitor ${\mathrm{HO}}^{•}$ radicals. HPLC analysis revealed that pCBA concentration decreased rapidly with ozone treatment time, and its concentration was considerably lower in ${\mathrm{Ce}}_{0.75}{\mathrm{Zr}}_{0.25}O_{2}$ ozonation than in single ozonation. This indicates that ${\mathrm{Ce}}_{0.75}{\mathrm{Zr}}_{0.25}O_{2}$ mixed oxide significantly promoted the decomposition of $O_{3}$ to ${\mathrm{HO}}^{•}$ radicals during the catalytic ozonation process. Their results also showed that ${\mathrm{BrO}}_{3}^{-}$ formation and $O_{3}$ decomposition was extremely rapid during the first 5 min of ozonation, further confirming that ${\mathrm{HO}}^{•}$ radicals play a major role during ${\mathrm{BrO}}_{3}^{-}$ formation. The organic compounds in water favours organic/${\mathrm{HO}}^{•}$ reactions more than ${\mathrm{Br}}^{-}$/${\mathrm{HO}}^{•}$ reactions, since the rate of reaction for oxidative degradation of organic compounds by ${\mathrm{HO}}^{•}$ radicals is faster than that for oxidizing ${\mathrm{Br}}^{-}$ by ${\mathrm{HO}}^{•}$ radicals [48]. Since the ${\mathrm{HO}}^{•}$ radicals facilitate the efficient degradation of organic substituents, therefore, the suppression of the oxidation of ${\mathrm{Br}}^{-}$ is favoured, leading to the minimization of ${\mathrm{BrO}}_{3}^{-}$ yield.

### 2.9. TiO_2_

Parrino et al. [49] investigated simultaneous ozonation and photocatalysis for purifying wastewater containing formic acid/4-nitrophenol and bromide ions. The initial ozonation experiments performed on formate and bromide ions in the presence and absence of TiO_2_, showed similar degradation rates, suggesting that reactions occurring on the TiO_2_ surface did not contribute to the degradation of the target compounds [50]. It was also observed that the oxidation of formate was not affected by the presence of bromide ion and the oxidation of bromide to bromate occurred only after the consumption of formate ions. Bromide ions reacted with hydroxyl radicals generated during photocatalysis, according to the following reaction scheme: (15)$${\mathrm{Br}}^{-}+{\mathrm{HO}}^{•}\;\to {\;\mathrm{Br}}^{•}+{\mathrm{OH}}^{-}$$
(16)$${\mathrm{Br}}^{•}+{\mathrm{HO}}^{•}\;\to \mathrm{HOBr}$$
(17)$$\mathrm{HOBr}\;\to {\;\mathrm{OBr}}^{-}+H^{+}$$
Lastly, the photoelectrons generated on the photocatalyst surface reduced the hypobromite species to bromide.
(18)$${\mathrm{OBr}}^{-}+2e^{-}+H_{2}O\;\to {\;\mathrm{Br}}^{-}+{\mathrm{OH}}^{-}$$

As illustrated, these pathways eventually lead to the recovery of bromide ion, Equation (18). Furthermore, if solution pH is in the range 6–8, a secondary pathway facilitates the conversion of hypobromous acid to bromide. The generated HOBr, as shown in Equation (16), primarily exists in its protonated form, and H_2_O_2_ generated during the photocatalytic reaction, acts as a scavenger for hypobromite, by reducing it to bromide [51].

From this outcome, they concluded that bromate generation can be prevented by interrupting the ozone treatment as soon as the oxidation of the organic species is almost complete. Furthermore, reducing bromate is also a more practical way to minimize its accumulation, and as per the previous reports, photocatalysis alone is efficient to convert bromate to bromide [51]. When 4-nitrophenol was substituted in the place of the formate ion, the formation of bromate, took place once again only after the disappearance of 4-nitrophenol, and was found to be faster than with formate ion. This implies that the type of organic contaminant in the water plays a decisive role in the amount of bromate formed.

### 2.10. β-FeOOH/Al_2_O_3_

Nie et al. [52] investigated bromate formation during the degradation of 2,4-dichlorophenoxyacetic acid (2,4-D) in ${\mathrm{Br}}^{-}$ containing water under uncatalysed and $\beta -{\mathrm{FeOOH}/\mathrm{Al}}_{2}O_{3}$ catalysed ozonation conditions. In uncatalysed ozonation, bromate yield increased rapidly to a maximum value of 21.5 ppb, but in $\beta -{\mathrm{FeOOH}/\mathrm{Al}}_{2}O_{3}$. ozonation, ${\mathrm{BrO}}_{3}^{-}$ formation was completely inhibited.

Furthermore, the experimental data showed that about 68% of ${\mathrm{BrO}}_{3}^{-}$ was adsorbed on the surface of$\;\beta -{\mathrm{FeOOH}/\mathrm{Al}}_{2}O_{3}$ during 2,4-D degradation, and with the addition of ozone, ${\mathrm{BrO}}_{3}^{-}$ was completely converted into ${\mathrm{Br}}^{-}$ within 180 min. In ozonation without 2,4-D, ${\mathrm{BrO}}_{3}^{-}$ was not reduced to$\;{\mathrm{Br}}^{-}$. However, ${\mathrm{BrO}}_{3}^{-}$ reduction was found to only occur with selected organic contaminants. 

Results also showed that no Fe^2+^ was formed when $\beta -{\mathrm{FeOOH}/\mathrm{Al}}_{2}O_{3}$ was present in water alone, however, a small amount of surface Fe^2+^ was observed when $\beta -{\mathrm{FeOOH}/\mathrm{Al}}_{2}O_{3}$ in water was ozonated. A further increase in surface Fe^2+^ was noticed when water in the presence of $\beta -{\mathrm{FeOOH}/\mathrm{Al}}_{2}O_{3}$ and 2,4-D was ozonated. The quantity of surface Fe^2+^ decreased rapidly when ${\mathrm{BrO}}_{3}^{-}$ was introduced, signifying that Fe^2+^ was responsible for ${\mathrm{BrO}}_{3}^{-}$ conversion to${\;\mathrm{Br}}^{-}$ [39,53]. The Fe^2+^ generated on $\beta -{\mathrm{FeOOH}/\mathrm{Al}}_{2}O_{3\;}$ arises from the reaction of Fe^3+^ with ${\mathrm{HO}}_{2}^{•-}{/O}_{2}^{•-}$, and the complexation of surface Fe^3+^ with the oxy-functional groups (-OH, -COOH). The organic pollutants or their oxygenated intermediates improves the reaction of Fe^3+^ with ${\mathrm{HO}}_{2}^{•-}{/O}_{2}^{•-}$, hence resulting in more surface Fe^2+^, which causes a higher ${\mathrm{BrO}}_{3}^{-}$ reduction rate.

### 2.11. Fe-Cu-MCM-41

Chen et al. [29] investigated the formation of bromate in the presence of Fe-Cu-MCM-41 during ozonation of ${\mathrm{Br}}^{-}$/Diclofenac containing water. They found that Fe-Cu-MCM-41 decreased the concentration of dissolved ozone, hence diminishing the direct reaction of O_3_ with${\;\mathrm{Br}}^{-}$. Ozonation of water containing only bromide ions, produced 276 ppb bromate, but in the presence of Fe-Cu-MCM-41 the bromate yield decreased to 151 ppb. Bromide in the presence of Diclofenac (DCF), saw a significant drop in bromate formation for both O_3_ alone and Fe-Cu-MCM-41/O_3_. During the initial treatment process, ${\mathrm{Br}}^{-}$ is oxidized to ${\mathrm{HBrO}/\mathrm{BrO}}^{-}\;$ and then to $\;{\mathrm{BrO}}_{3}^{-}$ under the action of O_3_ and $\;{\mathrm{HO}}^{•}$ radicals. The presence of DCF and its intermediates influences${\;\mathrm{BrO}}_{3}^{-}$ formation by competing with${\;\mathrm{Br}}^{-}$ and${\;\mathrm{HBrO}/\mathrm{BrO}}^{-}\;$ for O_3_ and $\;{\mathrm{HO}}^{•}$ radicals, thus inhibiting bromate formation. Also, the degradation of DCF decreases the solution pH to acidic, and bromate formation is not favoured in acidic medium [35]. 

### 2.12. Perovskite-Type Oxides, LaFeO_3_ and LaCoO_3_

Y. Zhang et al. [54] synthesized two perovskite-type oxides, LaFeO_3_ and LaCoO_3_, and examined their capacity to degrade benzotriazole (BZA) and minimize ${\mathrm{BrO}}_{3}^{-}$ formation in water during ozonation. The ozonation of an aqueous mixture of BZA and ${\mathrm{Br}}^{-}$ generated the most amount of ${\mathrm{BrO}}_{3}^{-}$. The bromate yield increased sharply for the first 20 min of ozonation and then showed a decreasing trend up to 120 min. The bromate yield decreased significantly after the addition of catalyst, especially during the first 30 min of ozonation, but the conversion of ${\;\mathrm{Br}}^{-}$ was faster with LaCoO_3_ compared with LaFeO_3_. The concentration of ${\mathrm{HBrO}/\mathrm{BrO}}^{-}$ was found to be higher in LaCoO_3_ ozonation than with LaFeO_3_, which explains its superior ${\mathrm{BrO}}_{3}^{-}$ minimization ability. The production of ${\mathrm{HO}}_{2}^{•-}/O_{2}^{•-}$ resulted in the generation of H_2_O_2_, which also contributed to the reduction of ${\;\mathrm{BrO}}_{3}^{-}$ to ${\mathrm{HBrO}/\mathrm{BrO}}^{-}$.

Y. Zhang et al. [54] further illustrated the reaction mechanism of LaFeO_3_ and LaCoO_3_ facilitated ozonation of benzotriazole (BZA) and$\;{\mathrm{BrO}}_{3}^{-}$ minimization. They concluded that LaFeO_3_ did not catalytically promote molecular ozone decomposition to reactive oxygen species (ROS), which is needed for BZA degradation, but instead rapidly reduced${\;\mathrm{BrO}}_{3}^{-}$. The reaction of${\;H}_{2}O_{2}$ over LaFeO_3_, suggested that ${\;H}_{2}O_{2}$ was used up in the presence of LaFeO_3_ and the consumed${\;H}_{2}O_{2}$ was not used to produce ${\mathrm{HO}}^{•}$ radicals. The $H_{2}O_{2}$ in [Fe-$H_{2}O_{2}$]_S_ more easily reduces ${\mathrm{BrO}}_{3}^{-}$ to $\mathrm{HOBr}/{\mathrm{OBr}}^{-}$.

On the other hand, LaCoO_3_ promoted the decomposition of ozone to ROS, which facilitated faster degradation of BZA and oxidation of ${\mathrm{Br}}^{-}$ to ${\mathrm{BrO}}_{3}^{-}$. Therefore, $\mathrm{HOBr}/{\mathrm{OBr}}^{-}$ concentration was lower in the presence of LaCoO_3_ than in ozonation alone. LaCoO_3_ accelerated the decomposition of BZA to $H_{2}O_{2}$. The $H_{2}O_{2}$ reduced ${\mathrm{BrO}}_{3}^{-}$ directly to form more $\mathrm{HOBr}/{\mathrm{OBr}}^{-}$. The cyclic reaction of Co^3+^/Co^2+^ also promoted BZA degradation and inhibition of ${\mathrm{BrO}}_{3}^{-}$ reduction. 

### 2.13. HZSM-5 Zeolites

T. Zhang et al. [55] studied the influence of H^+^-form high silica ZSM-5 (HZSM-5) zeolites with different Si/Al molar ratios (i.e., 25–300) on bromate formation. Their results showed that bromate yield increased with time in a single ozonation, O_3_/HZSM-5 and O_3_/CeO_2_. The bromate concentration in O_3_/HZSM-5 was significantly lower than in single ozonation and in O_3_/CeO_2_. The HZSM-5 with Si/Al ratios of 300 and 25 showed superior capacity for bromate minimization and reduced approximately 58% bromate formation potential after 20 min of ozone treatment, while CeO_2_ only reduced 22%. Further studies on HZSM-5 (Si/Al = 300) showed that its high efficiency for bromate minimization is related to its affinity to adsorb ${\mathrm{OBr}}^{-}$, a major intermediate in bromate formation [1]. The results have shown that HZSM-5 had no affinity to adsorb of ${\mathrm{Br}}^{-}$, ${\mathrm{BrO}}_{3}^{-}$ and HOBr, therefore no direct electron transfer reaction is expected on HZSM-5. However, the majority of ${\mathrm{OBr}}^{-}$ was rapidly adsorbed onto HZSM-5 within 0.5 min. They then concluded that the specific adsorption of ${\mathrm{OBr}}^{-}$ on the HZSM-5 prevents the oxidation of ${\mathrm{OBr}}^{-}$ to ${\mathrm{BrO}}_{3}^{-}$ in water. Their results also detected the presence of $H_{2}O_{2}$ in both single ozonation and ozonation with HZSM-5. Considerably higher yields of $H_{2}O_{2}$ were detected in single ozonation than in O_3_/HZSM-5 process, and the HZSM-5 neither adsorbed nor decomposed $H_{2}O_{2}$ in water. The lower $H_{2}O_{2}$ concentration in O_3_/HZSM-5 leads to lower bromate yields.

### 2.14. FeO_X_/CoO_X_

Gounden et al. [56], conducted a study on the degradation of hazardous halohydrin, 2,3-dibromopropan-1-ol (2,3-DBP) in water by ozonation alone and ozonation with Co loaded on Fe prepared by co-precipitation (Co-ppt) and a simple physical mixing method (Mixed). Their results showed that debromination of 2,3-DBP produced large quantities of ${\mathrm{Br}}^{-}$ and ${\mathrm{BrO}}_{3}^{-}$ ions. The Fe:Co (Mixed) catalyst was found to be more effective in suppressing the generation of bromate than the Fe:Co (Co-ppt) catalyst. The presence of Fe:Co (Mixed) lowered the solution pH from 6.8 to 5.7, which was an ideal condition for inhibiting bromate formation. The reaction pathway for conversion of ${\mathrm{Br}}^{-}$ to ${\mathrm{BrO}}_{3}^{-}$ was described in the presence of Fe-Co (Mixed) catalyst. Firstly, since pH of the initial solution (5.7), is higher than the pZc value (5.1) of the Fe-Co (Mixed) catalyst, its surface can comprise mostly of negative $\mathrm{Fe}-{\mathrm{Co}}^{-}$ active sites (Scheme 1). These sites repel the negatively charged bromide ions, thus preventing electron transfer reactions on the catalyst surface, resulting in a lower bromate yield. 

Secondly, since the pH of the initial solution is much lower than the pKa (8.8) of the HOBr/OBr^−^system, an equilibrium shift occurs to the left, thus favoring a higher yield of HOBr and lower ${\mathrm{OBr}}^{-}$. As ozone is more reactive towards HOBr than ${\mathrm{OBr}}^{-}$, a decrease in bromate yield is anticipated (Scheme 2).

A similar pattern of bromate formation, as illustrated in Scheme 3, was observed when 2,4,6-Tribromophenol (2,4,6-TBP) was ozonated with Fe-Co metal oxides. Using Fe-Co (Mixed) catalyst, only 5% of the available bromide was oxidized to bromate, whereas with Fe-Co (Co-ppt), 39% of bromide was converted. 

## 3. Factors Affecting Bromate Minimization

### 3.1. Effect of Initial Solution pH 

Previous studies have shown that lowering of pH to below 7, preceding ozonation results in a decrease in bromate formation [57]. A decrease of one pH-unit results in 50–63% reduction in ${\mathrm{BrO}}_{3}^{-}$ formation [58]. This decrease has been attributed to two factors: (i) At pH < 7, oxidized bromide is likely to primarily be found as hypobromous acid (HOBr), resulting in limited amounts of hypobromite $\left({\mathrm{OBr}}^{-}\right)$ available for reaction with ozone [18,59]:$$\mathrm{HOBr}⇌{\mathrm{OBr}}^{-}+H^{+}{\;\mathrm{pK}}_{a}=8.7$$

As the solution pH is increased, the concentration of ${\mathrm{OBr}}^{-}$ increases, hence promoting ${\mathrm{BrO}}_{3}^{-}$ production, since ${\mathrm{OBr}}^{-}$ is more reactive with ozone than HOBr [1]. The main oxidant for bromate formation in natural water is the hydroxyl radical. At a lower pH, the conversion of molecular ozone to hydroxyl radicals is low, therefore, the amount of bromate formed through the hydroxyl radical pathway is limited. At lower pH, the ratio of hydroxyl radical to ozone tends to be lower than at higher pH. The lowering of pH can also be problematic because it can result in poor or incomplete degradation of organic substrates, which can lead to the formation of various hazardous brominated organic compounds. Furthermore, for high alkalinity wastewaters, the lowering of pH is not economically feasible. 

Li et al. [19] studies confirmed that bromate formation increased significantly in ozonation alone as pH was increased from 6.3 to 9.5. This can be due to fact that in alkaline medium (i) ${\mathrm{OH}}^{-}$ shifts the acid/base equilibria of $\mathrm{HOBr}\;({\mathrm{pK}}_{a}=$ 8.8) towards ${\mathrm{OBr}}^{-}$, which reacts readily with both $O_{3}$ and ${\mathrm{HO}}^{•}$ [1], and (ii) ${\mathrm{OH}}^{-}$ decomposes $O_{3}$ to ${\mathrm{HO}}^{•}$ radicals, which enhances ${\mathrm{BrO}}_{3}^{-}$ formation. Their Ce_66_-MCM-48/O_3_ system minimized ${\mathrm{BrO}}_{3}^{-}$ formation and was also pH dependant. For pH range of 7.6–8.6, a higher minimization efficiency of 87–91% was attained by Ce_66_-MCM-48 after 10 min of ozonation. With a decrease in pH to 6.3, the inhibition efficiency decreased to 76%. When the pH was increased to 9.5, the minimization efficiency of Ce_66_-MCM-48 reduced to 82%. At high pH, ${\mathrm{OBr}}^{-}$ is the major species. It reacts rapidly with both $O_{3}$ and ${\mathrm{HO}}^{•}$ to form significant amounts of ${\mathrm{BrO}}_{3}^{-}$.

The experiments conducted by T. Zhang et al. [22] at controlled pH revealed that ${\mathrm{BrO}}_{3}^{-}$ yield increased rapidly in both single ozonation and in the $O_{3}{/\mathrm{CeO}}_{2}$ system as the pH was increased from 5.5 to 8.9. An 84% reduction in ${\mathrm{BrO}}_{3}^{-}$ yield was achieved at pH 6.2. They attributed the catalytic activity and ${\mathrm{BrO}}_{3}^{-}$ reduction to the surface charge of ${\mathrm{CeO}}_{2}$ and intermediary ${\mathrm{HOBr}/\mathrm{OBr}}^{-}$ speciation, which are pH dependent. When the pH of the solution is close to the ${\mathrm{pH}}_{\mathrm{pzc}}$ of ${\mathrm{CeO}}_{2}$ (6.6), its surface is not charged. If solution pH is below the ${\mathrm{pH}}_{\mathrm{pzc}}$ of ${\mathrm{CeO}}_{2}$ its surface becomes positively charged, due to protonation of its surface hydroxyl sites by water. This condition increases the proportion of HOBr, hence minimizing ${\mathrm{BrO}}_{3}^{-}$ formation. If solution pH is above the ${\mathrm{pH}}_{\mathrm{pzc}}$ of ${\mathrm{CeO}}_{2}$ its surface becomes negatively charged due to deprotonation of surface hydroxyl sites, thus continuously increasing the quantity of ${\mathrm{OBr}}^{-}$, which favours the formation of bromate ion.

Wu et al. [27] monitored$\;{\mathrm{BrO}}_{3}^{-}$ formation at different pH values during single ozonation and ozonation with nano$-{\mathrm{TiO}}_{2}$. Their results indicated that ozonation with nano$-{\mathrm{TiO}}_{2}$ favoured the formation of ${\mathrm{BrO}}_{3}^{-}$ as solution pH increased initially from 6.0 to 7.9. They also concluded that at high pH, rapid ozone decomposition is favoured, hence increasing production of hydroxyl radicals, resulting in higher ${\mathrm{BrO}}_{3}^{-}$ formation. A higher proportion of ${\mathrm{OBr}}^{-}$ is present at pH 7.9, which would also promote ${\mathrm{BrO}}_{3}^{-}$ formation. The increasing pH led to a slight decrease in ${\mathrm{BrO}}_{3}^{-}$ formation rate from 73.75% to 71.32%, displaying a reduced activity for nano$-{\mathrm{TiO}}_{2}$.

Xue et al. [31] observed that the initial solution pH had a significant influence on bromate formation during ozonation in the presence of Mn_100_-MCM-41(1 K min^−1^). The inhibition efficiencies for bromate formation were 96.7%, 83.4% and 68.2% at pH 6.5, 7.5 and 9.5 respectively. The increase in bromate formation with pH, is influenced by the equilibrium of ${\mathrm{HOBr}/\mathrm{OBr}}^{-}$ and the stability of ozone in aqueous media. The increasing pH favours the formation of more ${\mathrm{OBr}}^{-}$ ions, which readily decomposes $O_{3}$ to form ${\mathrm{HO}}^{•}$ radicals, therefore, accelerating bromate formation. In acidic conditions, ozone is stable and more HOBr is present, therefore, bromate formation is suppressed [60]. 

Chen et al. [33] observed that by increasing the initial solution pH from 3.0 to 9.0 increased bromate formation for both uncatalysed and Fe-Cu-MCM-41 catalysed ozonation, however, for the entire pH range Fe-Cu-MCM-41/O_3_ process generated lower bromate yield. As the pH increased to 9.0, bromate steadily accumulated, reaching 913 ppb in single ozonation and 335 ppb in Fe-Cu-MCM-41 ozonation. At the acidic condition, HBrO is favoured (pH < pK_a_), and since $O_{3\;}$ is more stable, fewer ${\mathrm{HO}}^{•}$ radicals are formed. As HBrO predominantly reacts with ${\mathrm{HO}}^{•}$, the oxidation pathway 2 in Figure 2 is suppressed and a reduced amount of bromate is formed. Under basic conditions, the equilibrium shifts towards ${\mathrm{BrO}}^{-}$, which is highly reactive towards both $O_{3}$ and ${\mathrm{HO}}^{•}$, resulting in accelerated bromate production [35].

Zhang et al. studied the influence of pH on bromate formation for the O_3_/HZSM-5 system [55]. In ozonation alone, it was observed that as solution pH increased from 6.6 to 9.3, the bromate yield increased rapidly from 4.9 ppb to 27 ppb. In catalytic ozonation with HZSM-5, the bromate yield increased more steadily from 2.8 ppb to 9.4 ppb. They attributed the drop in bromate formation to the adsorption of ${\mathrm{BrO}}^{-}$ on HZSM-5 at different pH levels. Considering the equilibrium constant of 10^-9^ for ${\mathrm{HOBr}/\mathrm{OBr}}^{-}$, the fraction of ${\mathrm{BrO}}^{-}$ in ${\mathrm{HOBr}/\mathrm{OBr}}^{-}$ at pH 8.0 and pH 9.3 is approximately 14% and 76%, respectively. This would mean that higher amounts of ${\mathrm{BrO}}^{-}$ can be adsorbed on HZSM-5 at pH 9.3 than at pH 8.0, so would the bromate reduction efficiency. However, their results have shown that the percent reduction in bromate formation increased only by 7.6%, when the solution pH was raised from 8.0 to 9.3. Since ${\mathrm{BrO}}^{-}$ is more reactive towards ozone than HOBr, and the ${\mathrm{HO}}^{•}{/\mathrm{OBr}}^{-}$ reaction rate is approximately two times that of ${\mathrm{HO}}^{•}/\mathrm{HOBr}$ [1]. Therefore, the increase in pH leads to a substantial increase in bromate yield in single ozonation. In HZSM-5 ozonation, O_3_ and ${\mathrm{HO}}^{•}$ compete with HZSM-5 for ${\mathrm{BrO}}^{-}$, thus resulting in lower bromate formation at higher pH. 

Kishimoto and Nakamura [61] concluded from their studies that hydroxyl radicals are more crucial than molecular ozone in bromate production. They demonstrated that in ozonation alone, ${\mathrm{BrO}}_{3}^{-}$ yield increased as ${\mathrm{Br}}^{-}$ concentration decreased at neutral pH in the absence of 4-chlorobenzoic acid (4-CBA). However, ${\mathrm{BrO}}_{3}^{-}$ yields considerably decreased compared to ${\mathrm{Br}}^{-}$ removal at acidic pH and in the presence of 4-CBA. Although acidic pH decreased ${\mathrm{BrO}}_{3}^{-}$ generation, it limited the oxidation capacity of ozone for successful 4-chlorobenzoic acid degradation. Therefore, the acidification during ozonation is favorable for${\;\mathrm{BrO}}_{3}^{-}$ minimization, but it has the disadvantage of affecting the removal efficiency of organic pollutants from water. 

### 3.2. Effect of Initial Bromide Concentration

Several studies have shown that the presence of small quantities of bromide ion can result in the generation of significant amounts of bromate ion during single ozonation. Bromate ion yield increased as bromide ion concentration increased. A few studies were conducted to investigate the influence of initial bromide concentration on the bromate formation during catalytic ozonation. 

Wu et al. [27] examined${\;\mathrm{BrO}}_{3}^{-}$ formation for various initial ${\mathrm{Br}}^{-}$ concentrations during single ozonation and ozonation with nano$-{\mathrm{TiO}}_{2}$. The data indicated that in single ozonation${\;\mathrm{BrO}}_{3}^{-}$ yield increased rapidly as a function of initial${\;\mathrm{Br}}^{-}$ concentration, however in ozonation with nano$-{\mathrm{TiO}}_{2}$ the ${\mathrm{BrO}}_{3}^{-}$ yield was significantly lower. When initial ${\mathrm{Br}}^{-}$ concentration increased from 0.4 ppm to 1.2 ppm, the reduction rate of ${\mathrm{BrO}}_{3}^{-}$ decreased from 67.22% to 47.11%, suggesting that the activity of nano$-{\mathrm{TiO}}_{2}$ is severely inhibited with an increase in initial${\;\mathrm{Br}}^{-}$ concentration.

The experiments conducted by T Zhang et al. [22] to study the influence of initial bromide concentration on bromate production showed that in single ozonation ${\mathrm{BrO}}_{3}^{-}\;$ yield increased rapidly from 0.5 ppm to 2 ppm, as the concentration of bromide ion increased. In ${\mathrm{CeO}}_{2}$ catalysed ozonation, ${\mathrm{BrO}}_{3}^{-}$ formation was significantly suppressed for ${\mathrm{Br}}^{-}$ concentrations ≤ 1.0 ppm, however, for ${\mathrm{Br}}^{-}$ concentrations > 1.0 ppm, $\;{\mathrm{BrO}}_{3}^{-}$ yield started to increase rapidly. The ${\mathrm{BrO}}_{3}^{-}$ yield in ${\mathrm{CeO}}_{2}$ catalysed ozonation was always lower than that obtained with uncatalysed ozonation. 

### 3.3. Effect of Ozone Dosage

Sufficient availability of ozone showed an increase in the bromate ion formation, until all bromide ion was converted to bromate ion [58]. von Gunten and Hoigne [18] have introduced a standard measure for the ozone concentration (C) as a function of reaction time (t), which is defined as the Ct value (mg/L·min) for ozone exposure. An increase in the quantity of ozone improves the Ct value during ozone treatment of water. Wu et al. [27] demonstrated that ${\mathrm{BrO}}_{3}^{-}$ yield kept on increasing as ozone concentration was increased in both single ozonation and nano$-{\mathrm{TiO}}_{2}$ ozonation, that is, for all experiments${\;\mathrm{BrO}}_{3}^{-}$ formation increased linearly as the ‘Ct value increased. When ozone dosage was increased from 2.22 ppm to 4.62 ppm, an improvement in the${\;\mathrm{BrO}}_{3}^{-}$ reduction rate from 62.94% to 75.66% was observed. The ${\mathrm{BrO}}_{3}^{-}$ formation rates in single ozonation were found to be much higher than in catalytic ozonation, however, no explanation was given for this trend.

Zhang et al. [55] showed that bromate yield increased rapidly from 7.8 ppb to 95 ppb in single ozonation as the ozone concentration was increased from 0.38 ppm to 1.16 ppm. In catalytic ozonation with HZSM-5, bromate yield increased much slower (from 4.3 ppb to 21 ppb) for the same increase in ozone dose. HZSM-5 may have depleted the concentrations of ozone and/or intermediate species, which are needed for bromate formation. 

### 3.4. Influence of Temperature Changes

The increasing temperature generally increases bromate ion production in water during ozonation. The effects of temperature are due to the following facts: (i) Ozone decomposition into ${\mathrm{HO}}^{•}$ radicals is favoured at higher temperatures; (ii) an increase in temperature enhances the reaction rate and (iii) the pK_a_ of the ${\mathrm{HOBr}/\mathrm{OBr}}^{-}$ system is temperature dependent.

The experimental data showing the influence of solution temperature on bromate minimization efficiency indicated that in the temperature range of 15℃ to 30℃ Ce_66_-MCM-48 catalytic ozonation showed nearly the same minimization efficiency as that of single ozonation [19]. This temperature-independent feature of Ce_66_-MCM-48 is advantageous for water treatment by ozonation.

The influence of solution temperature on ${\mathrm{BrO}}_{3}^{-}$ formation showed that, in single ozonation, the$\;{\mathrm{BrO}}_{3}^{-}$ yield increased moderately when the temperature was increased from 5 °C to 15 °C, and increased more sharply when raised from 15 °C to 25 °C. The generation of ${\;\mathrm{BrO}}_{3}^{-}$ in ${\mathrm{CeO}}_{2}$ ozonation was found to be similar to single ozonation, however much less${\;\mathrm{BrO}}_{3}^{-}$ was produced in${\;\mathrm{CeO}}_{2}$ ozonation [22]. 

### 3.5. Influence of Catalyst Dosage

Generally, the bromate yield increases as a function of catalyst dose. For example, bromate production with increasing nano$-{\mathrm{TiO}}_{2}$ dosage (0 to 200 ppm) investigated by Wu et al. [27] showed that when nano$-{\mathrm{TiO}}_{2}$ dose was increased from 0 to 100 ppm, the${\;\mathrm{BrO}}_{3}^{-}$ reduction rate increased from 0% to 72.59%. However, when nano$-{\mathrm{TiO}}_{2}$ dose increased from 100 to 200 ppm, the ${\;\mathrm{BrO}}_{3}^{-}$ reduction rate only went up to 74.27%. The nanoparticles have extremely high surface area, therefore, increasing nano$-{\mathrm{TiO}}_{2}$ dosage would result in more active catalytic sites for surface reactions. However, in aqueous solution, ozone concentrations are limited, hence the marginal increase in ${\mathrm{BrO}}_{3}^{-}$ reduction rate. 

## 4. Conclusions and Recommendations

The literature indicates that catalytic ozonation using appropriate catalyst materials is a better solution for bromate minimization than uncatalysed ozonation. However, there is still a need for more efficient and practically applicable catalysts to be explored for complete elimination of bromate formation during ozonation. All catalysts reported were able to significantly minimize BrO3− formation in comparison to ozonation alone, however, only few were able to minimize bromate formation below the 5 ppb limit. The following bromate inhibition strategies/mechanisms during catalytic ozonation of bromide containing waters were proposed:Increasing the number of hydroxyl groups on the catalyst surface resulted in enhanced ozone decomposition to HO• radicals, thus limiting the contribution of direct O3 for the sequential oxidation of Br−→HOBr/OBr−→ BrO3−. The formation of excess HO• is beneficial for removal of organic pollutants from the water.Redox reactions on the catalyst surface causes inhibition of  Br−→HOBr/OBr− and in some cases reduction of  BrO• to  HOBr/OBr−, thus limiting bromate formation. The lesser HOBr/OBr− concentration leads to lesser  BrO3−.The generation of hydrogen peroxide was detected in most catalytic ozonation systems, but was found to be lower than in ozonation alone. The lesser H2O2 means lesser HO• radicals, therefore, the oxidation rate of HOBr/OBr− to BrO• to BrO3− is diminished. Contrary to this, some authors observed an increase in H2O2, which they attributed to the reactive oxygen species, which are capable of consuming HOBr/OBr−. Further work on the relationship between H2O2 generation and bromate inhibition is therefore needed.The presence of phosphate and humic acid had a tendency to limit bromate formation, however, high levels of phosphate and humic acid can result in poor water quality.The limited studies on photocatalytic ozonation of bromide containing waters showed that the concentration of hypobromite species can be minimized by the photoelectrons generated on the photocatalyst surface, thus contributing to bromate reduction.Bromate reduction was enhanced in the presence of certain organic compounds, due to electron transfer reactions on the catalyst surface.Some catalysts have an affinity to adsorb critical intermediate species (OBr−) needed for bromate formation.Mixed metal oxides were found to effectively minimize bromate formation by simply lowering the initial solution pH to more acidic levels.

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
