# Peer review of "Advances in Treatment of Brominated Hydrocarbons by Heterogeneous Catalytic Ozonation and Bromate Minimization"

_molecules, 2019, doi:10.3390/molecules24193450_

Round 1

Reviewer 1 Report

The topic may be interesting for the readers of Molecules, although it is relatively narrow. Nevertheless, as a review, it contains (summarizes/surveys) numerous useful pieces of information. However, the presentation of them seems to be a bit fuzzy. For instance, already in the Introduction Fig. 1, which ought to be an important scheme to give the skeleton/structure of the subsequent parts, is rather ambiguous, besides, it is not in full accordance with the text regarding it (lines 50-58). There are considerably number of errors of grammar  and composition in the text, e.g. singular-plural problems (e.g.,in lines 30-31, 51), missing commas at many places, missing verb (in line53), comma instead of period (line 72), superfluous words/prepositions (e.g., in lines 73, 103, 311, 414, 443), etc. It needs a thorough improvement in this respect. Also the usage of letters is rather sloppy. The size in lines 320-321 is too big. The letters are italics in lines 262-272.

Regarding the discussion of the various catalysts, it ought to compare them in a more explicit way, emphasizing both the advantages and the drawbacks.

Author Response

All corrections suggested by reviewers are implemented and highlighted in red in the revised manuscript

Reviewer 1

The topic may be interesting for the readers of Molecules, although it is relatively narrow. Nevertheless, as a review, it contains (summarizes/surveys) numerous useful pieces of information.

The topic is relatively narrow as this review focused on bromate minimization strategies during heterogeneous catalytic ozonation of waters containing brominated hydrocarbons. The literature survey revealed that only few studies were conducted on this topic and there is certainly potential for future work. 

However, the presentation of them seems to be a bit fuzzy. For instance, already in the Introduction Fig. 1, which ought to be an important scheme to give the skeleton/structure of the subsequent parts, is rather ambiguous, besides, it is not in full accordance with the text regarding it (lines 50-58).

The paragraph in question was re-structured as shown below for more clarity. It is included in the text (Lines 48 – 58).

In aqueous systems, ozone oxidizes bromide to bromate via three different pathways [14]. Dominance of a particular pathway is dependent on the amount of bromide, organic carbon and pH of the substrate solution. As illustrated in Figure 1, the first pathway (direct pathway) is initiated by the reaction of bromide ion with molecular ozone to form. The is further oxidized by dissolved  to and finally to . The second pathway (direct/indirect pathway) is facilitated by molecular ozone, resulting in the formation of . However, in this route the formed is oxidized byradicals to a series of highly reactive oxygenated radicals. Further ozonation produces  ions. According to Richardson et al. [15] this pathway is favoured if solution pH and alkalinity of the water is high. In the third pathway the  radicals interact with bromide ions resulting in the generation of  radicals, which disproportionate to bromite ions. The bromite ions are then oxidized by molecular ozone to produce bromate ions.

There are considerably number of errors of grammar and composition in the text, e.g. singular-plural problems (e.g., in lines 30-31, 51), missing commas at many places, missing verb (in line53), comma instead of period (line 72), superfluous words/prepositions (e.g., in lines 73, 103, 311, 414, 443), etc. It needs a thorough improvement in this respect.

The entire manuscript was checked and corrected for errors with respect to grammar, and the sentences were redrafted for clarity. Changes are highlighted in red in the text. Specific suggested changes (shown below) are also effected.

Lines 30-31 and 51 corrected in the text.

Line 53 corrected in the text.

Line 72 corrected in the text.

Lines 73, 103, 311, 414, 443 corrected in the text.

Also the usage of letters is rather sloppy. The size in lines 320-321 is too big. The letters are italics in lines 262-272.

The font of letters in lines 320-321 is corrected.

The italic letters in lines 262-272 are changed in the text.

All the changes to the text are indicated in red.

Regarding the discussion of the various catalysts, it ought to compare them in a more explicit way, emphasizing both the advantages and the drawbacks.

The advantages and limitations of each catalyst reviewed are outlined in the text. Some of the sentences in the text were reworded to emphasise this aspect. Their catalytic action/mechanism are summarised in the conclusion section.

Reviewer 2 Report

 On account of the manuscript Molecules-592772, entitled “Advances in Treatment of Brominated Hydrocarbons by Catalytic Ozonation and Bromate Minimization” by Asogan N. Gounden and Sreekanth B. Jonnalagadda, the authors conducted systematic review for recent advances on bromate reduction in aqueous media by catalytic ozonation for creating a platform for future research and a quest to find environment friendly and efficacious catalysts. The topic is important to conduct environmental risk management related to catalytic ozonation, and the authors got interesting results. The manuscript was well written and designed. After careful consideration, I made a decision that the manuscript is acceptable for publication in its present form.

Author Response

On account of the manuscript Molecules-592772, entitled “Advances in Treatment of Brominated Hydrocarbons by Catalytic Ozonation and Bromate Minimization” by Asogan N. Gounden and Sreekanth B. Jonnalagadda, the authors conducted systematic review for recent advances on bromate reduction in aqueous media by catalytic ozonation for creating a platform for future research and a quest to find environment friendly and efficacious catalysts. The topic is important to conduct environmental risk management related to catalytic ozonation, and the authors got interesting results. The manuscript was well written and designed. After careful consideration, I made a decision that the manuscript is acceptable for publication in its present form.

Response: No comments were made by reviewer 2 that required corrections.

Reviewer 3 Report

The critical comments are as follows:

1.     Throughout the manuscript, the level of English used is not up to the standard of the journal. The sentences are long and badly worded with repetitive words. Please consider breaking longer sentences into smaller fragments for easy understanding. Authors are advised to seek help from a native English speaker. For instance, L10: “during in use of ozone”

2.     L15: “most of these methods had a negative effect”, what are those? Enlist some.

3.     Data from the third party such as reprinted Figures and Tables should be clearly marked with credit statement such as “Reproduced/Adapted/Reprinted from Ref. [X] with permission from a respective source publisher. Copyright (YEAR) Publisher.

4.     L262-272: why italics?

5.     What was the methodology of the review? What was the literature inclusion/exclusion criteria? There is a plethora of literature available, why authors have only selected these ones?

6.     Referencing is not right and consistent. Above points are some highlighted areas. This should be improved as there are many reports available from the year 2018-2019.

7. Editorial issues: The Latin names and Greek letters should be presented in italic in the whole manuscript, units presentation should be unified in the whole manuscript, abbreviations presentation should be unified.

Author Response

Reviewer 3

Throughout the manuscript, the level of English used is not up to the standard of the journal. The sentences are long and badly worded with repetitive words. Please consider breaking longer sentences into smaller fragments for easy understanding. Authors are advised to seek help from a native English speaker. For instance, L10: “during in use of ozone”

The manuscript was thoroughly checked and attempt is made to improve English to meet the standards of the journal. Help from a competent 3rd party English speaker was also sought.

 A number of longer sentences in the text were broken up into smaller sentences for clarity. Badly worded sentences are redrafted.

L10 is corrected in the text.

All changes/corrections are reflected in red in the text.

L15: “most of these methods had a negative effect”, what are those? Enlist some.

The sentence in L15 is changed to “However, most of the above strategies had a negative effect on the ozonation efficiency” to give more clarity.

The strategies/methods are listed in L13-14 of the abstract.

Data from the third party such as reprinted Figures and Tables should be clearly marked with credit statement such as “Reproduced/Adapted/Reprinted from Ref. [X] with permission from a respective source publisher. Copyright (YEAR) Publisher.

As requested, the following changes are made to the Figure Captions and included in the text:

Figure 1. Bromate formation pathways reproduced from Ref. [14], with permission from Journal of American Water Works Association (1997)

Figure 2. Formation route for bromate reprinted from Ref. [33], with permission from Journal of Hazardous Materials (2018)

L262-272: why italics?

The italic letters in lines 262-272 are changed in the text for consistency.

What was the methodology of the review? What was the literature inclusion/exclusion criteria? There is a plethora of literature available, why authors have only selected these ones?

The review focused on bromate minimization strategies during heterogeneous catalytic ozonation of waters containing brominated hydrocarbons. The make this more explicit, the word “HETEROGENEOUS” was added to the title. Therefore, only those pertaining to heterogeneous catalytic ozonation was selected for the review.  The literature survey revealed that few studies were conducted on this topic and there is certainly potential for future work.

Referencing is not right and consistent. Above points are some highlighted areas. This should be improved as there are many reports available from the year 2018-2019.

An open literature survey was conducted on this topic up to and including recent work done in 2019. All information reported with respect to this topic was included in this review.

Editorial issues: The Latin names and Greek letters should be presented in italic in the whole manuscript, units presentation should be unified in the whole manuscript, abbreviations presentation should be unified.

The above editorial issues were checked and were necessary corrections were effected.

Round 2

Reviewer 1 Report

The manuscript has been substantially enhanced, although there remain some faults, or new ones have been generated.

For instance, in line 59, the sentence states the production of BrO as a result of the reaction between bromide and OH radical, while it is formed in the reaction betwween Br and ozone. New errors of grammar have been generated e.g. in line 17 ( ..processes .....has..) or lines 309-310 (Since......, therefore....). Besides, the letters have been written in red at many places, although the text has not been changed there at all, e.g much less changes have been made than indicated by the red letters.

Reviewer 3 Report

The revised version reads well. Authors have addressed all the comments raised in the last review. This manuscript can now be accepted for publication.